# Effect of the Coulomb repulsion and oxygen level on charge distribution and superconductivity in the Emery model for cuprates superconductors.

Louis-Bernard St-Cyr[1] and David Sénéchal[1]

[1]*Département de physique and Institut quantique, Université de Sherbrooke, Sherbrooke, Québec, Canada J1K 2R1*
(Dated: March 12, 2025)

The Emery model (aka the three-band Hubbard model) offers a simplified description of the copper-oxide planes that form the building blocks of high-temperature superconductors. By contrast with the even simpler one-band Hubbard model, it differentiates between copper and oxygen orbitals and thus between oxygen occupation ($n_p$) and copper occupation ($n_d$). Here we demonstrate, using cluster dynamical mean field theory, how the two occupations are related to the on-site Coulomb repulsion $U$ on the copper orbital and to the energy difference $\epsilon_p$ between oxygen and copper orbitals. Since the occupations ($n_p$ and $n_d$) have been estimated from NMR for a few materials (LCO, YBCO and NCCO), this allows us to estimate the value of $U - \epsilon_p$ for these materials, within this model. We compute the density of states for these and the effect of ($U, \epsilon_p$) on the $n_d$-$n_p$ curve, superconductivity, and antiferromagnetism.

## I. INTRODUCTION

High-temperature superconductors, nearly forty years after their discovery, still constitute a theoretical challenge. The dozens of materials in this category all share a common feature: planes of copper oxide (CuO$_2$). Thus one would naturally expect a universal explanation of superconductivity in cuprates based on a model of these planes only. The one-band Hubbard model has long been the focus of research in this direction, in part because of its relative simplicity: a few hopping parameters, the Coulomb repulsion $U$ and the chemical potential controlling the electron density. That model features a single band crossing the Fermi level, argued to be a combination of copper $d_{x^2-y^2}$ and oxygen $p_x$ and $p_y$ orbitals, forming excitations above what we call the Zhang-Rice singlet [1]. Various theoretical approaches applied to this model have provided evidence for a dome of $d$-wave superconductivity away from half-filling, in addition to a robust antiferromagnetic phase close to half-filling [2–6], although there is also evidence, in some parameter range, that superconductivity is not the lowest-energy state [7, 8].

A more realistic description of the CuO$_2$ planes is provided by the Emery-VSA (Varma-Schmitt-Rink-Abrahams) model [9, 10], which involves the copper $d_{x^2-y^2}$ orbital and two oxygen $p$-orbitals per unit cell. Within that model, the question of the distribution of holes between the copper and oxygen orbitals now has meaning. Since the occupation of both orbitals can be estimated from nuclear magnetic resonance (NMR) experiments [11], a comparison with NMR data can in principle be used to impose constraints on some of the parameters of that model, namely the on-site Coulomb interaction on the copper orbitals and the energy levels of both orbitals. This is what this paper is mainly about. This will also lead us to revisit the relation of superconductivity with $U$ and the matter of localized vs itinerant antiferromagnetism in NCCO.

This paper is organized as follow: In Sect. II we review the definition of the Emery model and the method used (Cluster dynamical mean-field theory or CDMFT). The latter is mostly discussed in the Appendix. In Sect. III we

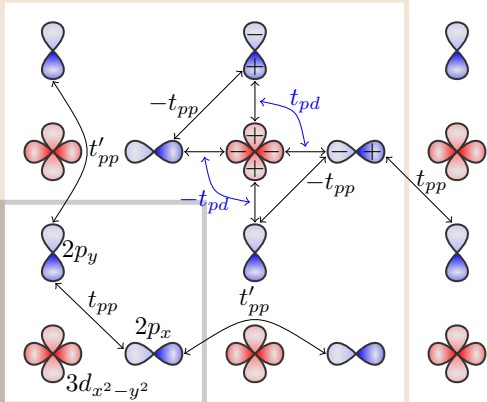

Figure 1. Schematic representation of the Emery model used in this work: Copper $d_{x^2-y^2}$ orbitals are shown in red and Oxygen $p_x, p_y$ orbitals in blue. Nonzero hopping parameters are indicated. The gray square defines the unit cell, and the beige square the super unit cell used in CDMFT (see appendix).

present our numerical results for the relative hole and electron distribution among the Cu and O orbitals as a function of doping, interaction strength $U$ and oxygen energy level $\epsilon_p$. We also show the effect of these parameters on the superconducting order parameter at zero temperature and we compare the charge-distribution curves with those of real materials obtained from NMR. Finally in Sect. IV we discuss the applicability of the Emery model to these materials and conclude.

## II. MODEL AND METHOD

### A. The Emery model

Let us first review the basics of the Emery model. It is schematically illustrated in Fig. 1. In second-quantized language, the kinetic energy $H_{\text{kin.}}$ and interaction $H_{\text{int.}}$ are ex-

pressed as follows:

$$H_{\text{kin.}} = \sum_{\mathbf{r},\sigma} \left\{ (\epsilon_p - \mu)(n_{\mathbf{r}+\mathbf{a}_x} + n_{\mathbf{r}+\mathbf{a}_y}) - \mu n_{\mathbf{r}} \right.$$
$$+ t_{pd} \sum_{\mathbf{r}} \sum_{\mathbf{a}=\mathbf{a}_x,\mathbf{a}_y} \sum_{\nu=\pm 1} \nu d_{\mathbf{r},\sigma}^\dagger p_{\mathbf{r}+\nu\mathbf{a},\sigma} + \text{H.c}$$
$$+ t_{pp} \sum_{\mathbf{r}} \sum_{\nu,\nu'=\pm 1} \nu\nu' p_{\mathbf{r}+\nu\mathbf{a}_x,\sigma}^\dagger p_{\mathbf{r}+\nu'\mathbf{a}_y,\sigma} + \text{H.c}$$
$$\left. + t'_{pp} \sum_{\mathbf{r}} \sum_{\mathbf{a}=\mathbf{a}_x,\mathbf{a}_y} p_{\mathbf{r}-\mathbf{a},\sigma}^\dagger p_{\mathbf{r}+\mathbf{a},\sigma} + \text{H.c} \right\} \qquad (1)$$

$$H_{\text{int.}} = U \sum_{\mathbf{r}} n_{\mathbf{r}\uparrow} n_{\mathbf{r}\downarrow} \ , \qquad (2)$$

where $\mathbf{r}$ labels the position of the copper atoms and $2\mathbf{a}_x = 2\mathbf{a}_y$ are the Bravais square lattice vectors. The operator $d_{\mathbf{r},\sigma}$ annihilates an electron of spin $\sigma$ on the copper $3d_{x^2-y^2}$ orbital located at the position $\mathbf{r}$. The operator $p_{\mathbf{r}+\mathbf{a},\sigma}$ annihilates an electron of spin $\sigma$ on the oxygen $2p_x$ or $2p_y$ orbital located at $\mathbf{r} + \mathbf{a}_{x,y}$. $n_{\mathbf{r},\sigma} = d_{\mathbf{r},\sigma}^\dagger d_{\mathbf{r},\sigma}$ is the number of electrons of spin $\sigma$ in the Cu orbital at site $\mathbf{r}$, whereas $n_{\mathbf{r}+\mathbf{a}_x}$ and $n_{\mathbf{r}+\mathbf{a}_y}$ are the number of electrons (summed over spins) on each of the oxygen orbitals. $\epsilon_p$ represents the difference

in energy between the oxygen and copper orbital and the energy $\epsilon_d$ of the copper orbital is used as reference. $t_{pd}$ is the first-neighbor hopping amplitude between a copper and the oxygen orbitals that are directly around it. $t_{pp}$ is the hopping amplitude between two nearest-neighbor oxygen orbitals (diagonally) and $t'_{pp}$ is the third-neighbor hopping amplitude between two oxygen orbitals over a copper orbital. The sign convention for the hopping parameters used in this work is illustrated in Fig. 1. $U$ represents the energy cost of double occupancy caused by the Coulomb repulsion between electrons located on the same copper orbital. The corresponding local repulsion $U_p$ on the oxygen orbitals is neglected because of its smaller value, as justified by density functional theory (DFT)[12], and also because the oxygen orbitals are nearly filled in the cases of interest. The chemical potential $\mu$ allows to control the filling (electron or hole doping) of the system.

We can express the kinetic part $H_{\text{kin.}}$ more compactly in reciprocal space:

$$H_{\text{kin.}} = \sum_{\mathbf{k}\sigma} \Phi_{\mathbf{k},\sigma}^\dagger \mathbf{h_k} \Phi_{\mathbf{k},\sigma} \qquad (3)$$

where the different annihilation operators are assembled in the multiplet $\Phi_{\mathbf{k},\sigma} = (d_{\mathbf{k}\sigma}, p_{\mathbf{k}\sigma}^x, p_{\mathbf{k}\sigma}^y)$ and the $3 \times 3$ matrix $\mathbf{h_k}$ is

$$\mathbf{h_k} = \begin{pmatrix} 0 & t_{pd}(1-e^{-ik_x}) & t_{pd}(1-e^{-ik_y}) \\ t_{pd}(1-e^{ik_x}) & \epsilon_p + 2t'_{pp}\cos k_x & t_{pp}(1-e^{ik_x})(1-e^{-ik_y}) \\ t_{pd}(1-e^{ik_y}) & t_{pp}(1-e^{-ik_x})(1-e^{ik_y}) & \epsilon_p + 2t'_{pp}\cos k_y \end{pmatrix}. \qquad (4)$$

Note that we did not renormalize $\epsilon_p$ by including the $-2t_{pp}$ contribution of the oxygen orbital energy, in contrast with Ref. [13].

Since the unit cell contains three orbitals, the maximum occupation is $n = 6$. The "half-filling" state is set to five electrons per unit cell, with the O orbitals being fully filled and the Cu orbital half-filled, in the zero-hopping limit. At finite hopping the O orbitals are slightly less than fully filled and the Cu orbital is slightly more than half-filled [11]. Such an undoped state describes a charge transfer insulator (CTI) when the interaction $U$ is large enough to split the Cu energy band into an upper (UHB) and lower (LHB) Hubbard band [14]. At those large $U$ values, the UHB is pushed past the energy of the oxygen band, also called the charge-transfer band (CTB). Partial holes on the O orbitals cause a loss of electronic weight in the CTB and give rise to the Zhang-Rice singlet band (ZRB) [1] made of low-energy states formed by the mixing of Cu and neighboring O orbitals. In a CTI, an insulating gap, called a charge transfer gap (CTG), is formed between the ZRB and the UHB upon increasing $U$. This CTG in the Emery model is the analog of the Mott gap in a Mott insulator (MI). The different spectral features of the Emery model are illustrated in the density of

states of Fig. 6a below.

Finally, a note on *double counting*. The relative energy level $\epsilon_p$ between O and Cu orbitals computed from DFT is subject to a renormalization when the Coulomb interaction $U$ is introduced. Indeed, Coulomb interactions are already taken into account at the DFT level and impact the value of $\epsilon_p$. When $U$ is introduced in an explicit way in the Emery model, the DFT effect of that local Coulomb interaction should be subtracted in order to avoid counting it twice. This of course is rather difficult to evaluate exactly. A rough estimate of this double counting can be obtained by computing the effect of $U$ at the mean-field (Hartree) level, i.e., by replacing

$$U n_{\mathbf{r}\uparrow} n_{\mathbf{r}\downarrow} \quad \text{by} \quad U\left(\bar{n}_{\mathbf{r}\uparrow} n_{\mathbf{r}\downarrow} + n_{\mathbf{r}\uparrow}\bar{n}_{\mathbf{r}\downarrow} - \bar{n}_{\mathbf{r}\uparrow}\bar{n}_{\mathbf{r}\downarrow}\right) \ , \qquad (5)$$

where $\bar{n}_{\mathbf{r}\sigma}$ is the occupation of the Cu orbital of spin $\sigma$ at site $\mathbf{r}$. In the paramagnetic state, $\bar{n}_{\mathbf{r}\uparrow} = \bar{n}_{\mathbf{r}\downarrow}$ and this brings a contribution $\frac{1}{2}U\bar{n}_{\mathbf{r}}$ to the copper orbital energy $\epsilon_d$ that must be subtracted, which leads to a corresponding correction $\frac{1}{2}U\bar{n}_{\mathbf{r}}$ to $\epsilon_p$ since $\epsilon_d$ is used as the reference energy. Such a correction was already applied in Refs [15, 16] but manifestly depends on $U$. Hence one cannot set $U$ in the model without also playing with $\epsilon_p$. This is why in the following we

study both the effect of $U$ and $\epsilon_p$ on the properties of the model.

## B. Cluster dynamical mean-field theory

In this work we approximately solve the Emery model using cluster dynamical mean field theory (CDMFT) [17, 18] with an exact diagonalization (ED) impurity solver at zero temperature, implemented in the open-source library Pyqcm [19]. CDMFT is an extension of dynamical mean field theory (DMFT). DMFT is an embedding procedure that incorporates the correlations of a single correlated orbital with its lattice environment into a hybridization of that orbital with a set of non-interacting orbitals (the "bath"). DMFT assumes that the self-energy is purely local, i.e., its momentum dependence is neglected. To include the latter, which is essential to describe $d_{x^2-y^2}$ superconductivity in cuprates, we need to upgrade the single-site impurity problem (DMFT) into a cluster impurity problem (CDMFT), in which a four-site cluster of Cu atoms is embedded into an effective environment. In this work the decomposition of the infinite lattice into clusters is done in the same way as in Ref. [20]: the lattice is tiled with identical units, each of which consisting of a cluster of $N_d = 4$ correlated Cu sites and another, overlapping cluster of $N_O = 8$ uncorrelated oxygen sites, as shown in Fig. 1. The correlated Cu cluster is augmented by an 8-orbital bath that represents that cluster's environment, and defines an "impurity problem". See Appendix A for technical details.

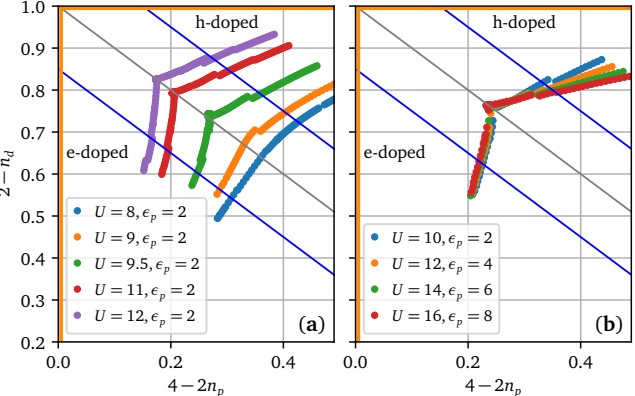

Figure 2. Relative filling $n_d$ of copper and $n_p$ of oxygen in a model for Bi$_2$Sr$_2$Ca2Cu$_3$O$_{10}$. Panel (a): the effect of $U$ for a fixed value of $\epsilon_p$. Panel (b): the effect of $U$ when the nominal gap $U - \epsilon_p$ is kept constant. The gray line sits at half-filling and the blue lines at ±15% doping. The orange lines show the zero-hopping limit.

Table I. Hopping parameters used in this work, from Refs [15, 16].

| Material | $t_{pd}/t_{pp}$ | $t'_{pp}/t_{pp}$ | $(\epsilon_p - \epsilon_d)/t_{pp}$ |
|---|---|---|---|
| BSSCO | 2.139 | 0.258 | 2.01 |
| LSCO | 2.172 | 0.1609 | 4.08 |
| YBCO | 1.902 | 0.223 | 3.05 |
| NCCO | 2.148 | 0.389 | 2.98 |

## III. RESULTS

### A. Doping curves

The effect of $U$ and $\epsilon_p$ on the distribution of holes and electrons in the Emery model can be seen by plotting the occupation of O orbitals against that of the Cu orbital. We have computed the distribution of electrons and holes for a range of values of $U$ and $\epsilon_p$ in a model of Bi$_2$Sr$_2$Ca$_2$Cu$_3$O$_{10}$ (BSCCO) as shown in Fig. 2. The hopping parameters are taken from ref. [15]. Note that we also obtained solutions on the "wrong" side of half-filling for this material (and for others in this paper), i.e, for both hole and electron doping, irrespective of its actual hole- or electron-doped character.

Let $n_d$ ($n_p$) denote the average occupation of copper (oxygen) orbitals. In Fig. 2, one notices a discontinuity in the slope $\frac{1}{2} dn_d/dn_p$ of the curves at half-filling, i.e., along the diagonal gray line defined by $n_d + 2n_p = 5$. This discontinuity occurs for all curves with $U > 9$ on the figure and is a signature of the charge transfer gap (CTG). On the hole-doped side, the smaller slope indicates that the hole density increases faster on the O orbitals than on the Cu orbital when removing electrons. By contrast, the slope on the electron-doped side indicates that the doped electrons mostly go to the Cu orbital. Note from Fig. 2a that the slope discontinuity between the electron- and hole-doped sides of the curve increases with $U$, that is, as $U$ gets stronger, the doped hole (electrons) go increasingly towards the O (Cu) orbitals. Both of these observations are in agreement with experimental data [21, 22] (see Fig. 5) and with theoretical calculations using a different impurity solver [23, 24]. In the atomic limit, i.e., when setting all hopping terms to zero (but keeping $\epsilon_p$), the corresponding curve would follow the $n_p = 2$ axis on the electron-doped side, and the $n_d = 1$ axis on the hole-doped side (orange lines on the figure). At lower values of $U$, crossing the half-filling axis has no effect on the slope of the curve, corresponding to a continuously hybridized Cu-O band at the Fermi level.

In the zero-hopping limit, the CTG gap is precisely $\Delta_n = U - \epsilon_p$. We will call this the *nominal gap*, as opposed to the real CTG gap obtained from the computed density of states when hopping is present. The actual CTG is much smaller than $U - \epsilon_p$ because of the finite bandwidth in the presence of hopping. From Fig. 2b, one sees that increasing $\epsilon_p$ while keeping the nominal gap $U - \epsilon_p$ constant barely affects the position of the curve along the half-filling line and makes the curves flatter on the hole-doped side while having no effect on the electron-doped side. Indeed, increasing

$\epsilon_p$ pushes the lower Hubbard band to even lower energies, thus making the charge transfer band even more dominated by the oxygens and removing copper content from the ZRS band. Hence added holes go more and more on the oxygen atoms. On the other hand, that increase has apparently much less effect on the oxygen content of the upper Hubbard band, because the relative location of the UHB and the CTB remains roughly the same, and so does the actual CTG. We conclude from this that the ZRS band gets its Cu content from both the lower and upper Hubbard bands.

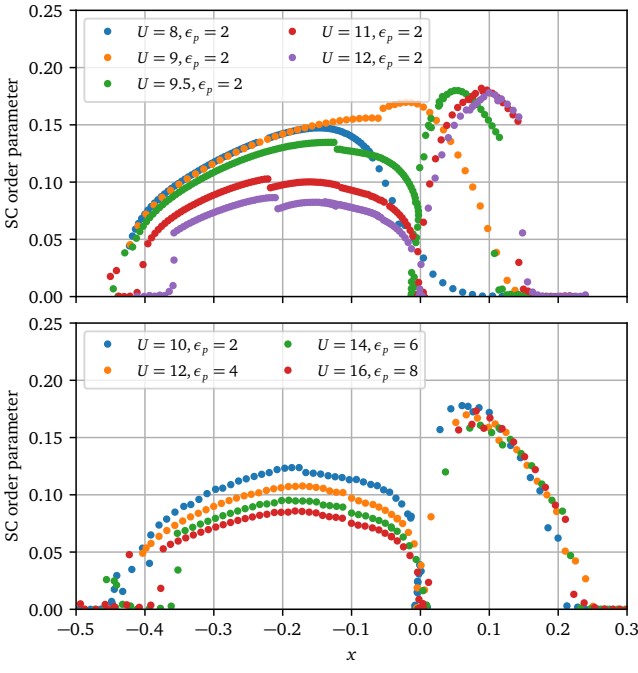

Figure 3. $d$-wave superconducting order parameter vs electron-doping $x$ in a models for BSCCO with varying $U$ at fixed $\epsilon_p$ (top) and varying $U$ at constant CTG, i.e., varying $\epsilon_p$ (bottom).

### B. Superconductivity

Fig. 3 shows the $d$-wave superconducting order parameter $\Psi$, the average of the pairing operator (A13), as a function of electron doping $x$ (negative for hole doping) for the same values of $U$ and $\epsilon_p$ as in Fig. 2. On the top panel, $U$ is varied while $\epsilon_p$ is kept constant, whereas in the bottom panel $U$ is varied while $U - \epsilon_p$ is kept constant. We first note that superconductivity exists even when the CTG vanishes ($U = 8$ and $U = 9$). It is even at its maximum at half-filling just under the Mott transition ($U = 9$), but shifts towards the hole-doped region as $U$ is lowered to $U = 8$. Note that these calculations ignore the presence of antiferromagnetism near half-filling. In all cases, the maximum order parameter $\Psi_{max}$ on the electron-doped side is not affected by $U$ or $\epsilon_p$, whereas it decreases with $U$ on the hole-doped side, although less markedly if the nominal gap $U - \epsilon_p$ is kept constant.

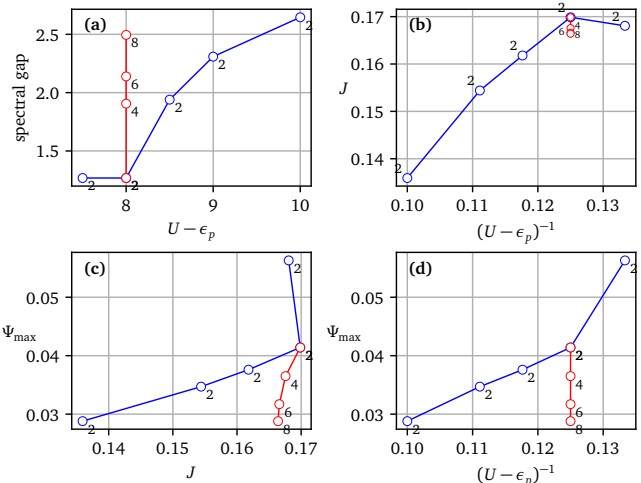

Figure 4. Various properties of the Emery model computed from CDMFT and the parameters used for BSCCO in Fig. 2. Panel (a) shows the CTG computed from the density of states, as a function of $U - \epsilon_p$. Panel (b) shows the superexchange estimate $J$ as a function of $(U - \epsilon_p)^{-1}$. Panel (c) shows the maximum superconducting order parameter on the hole-doped side (see Fig. 3) as a function of the superexchange $J$ at half-filling, and Panel (d) the same quantity as a function of $(U - \epsilon_p)^{-1}$. The value of $\epsilon_p$ is written next to each data point.

In Ref. [20] it was argued that the main factor influencing $\Psi_{max}$ was the effective super-exchange $J$ computed at half-filling. This quantity is computed on the impurity (the four-site plaquette) and is the position of the first (dominant) pole in the dynamic antiferromagnetic spin susceptibility as a function of frequency [20, 25]. Fig. 4b shows how $J$ depends on the inverse nominal gap, whereas Fig. 4c shows how $\Psi_{max}$ is related to $J$. As can be seen, the relation between the two is roughly linear for fixed $\epsilon_p = 2$ until the CTG almost closes. The linear relation also holds as a function of the inverse nominal gap (Fig. 4d). Note that the data shown in red, with constant nominal gap, corresponds to a single value of $J$, but a sizeable variation of $\Psi_{max}$, demonstrating that $J$ is not the only factor at play.

Interestingly, all curves of $\Psi$ with a nonzero CTG also show a discontinuity in the hole-doped region, that appears at optimal doping, as illustrated in Fig. 3. This discontinuity within the superconducting phase was also observed in Refs [20, 26] and marks the boundary between a superconducting phase with and without pseudogap.

### C. Comparison with NMR

We studied three different cuprate compounds and computed the hole distribution between oxygen and copper sites in the superconducting phase, with the goal of estimating the strengths of the Coulomb interaction $U$ and of the orbital energy $\epsilon_p$ in those materials. We looked at two hole-doped cuprates, $La_2CuO_4$ (LCO) and $YBa_2Cu_3O_7$ (YBCO),

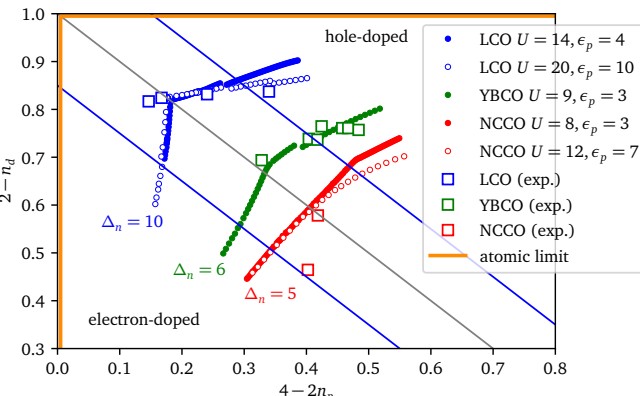

Figure 5. Relative filling $n_d$ of Cu $n_p$ of O orbitals. NMR data from [11] (squares) compared with our CDMFT results for LCO, YBCO and NCCO. The orange lines are the zero-hopping limit of the Emery model. The values of the nominal gap $\Delta_n = U - \epsilon_p$ are indicated.

using the hopping parameters from Ref. [15] given in Table I, and varying also $\epsilon_p$ from the values of that table. We also looked at one electron-doped cuprate, $Nd_2CuO_4$ (NCCO), using the band parameters from Ref. [16]. We compare in Fig. 5 the hole distributions of these three materials computed from CDMFT with the nuclear magnetic resonance (NMR) data from Ref. [11]. For each of the three materials, we computed the doping curves for many values of $U$ and the value of $\epsilon_p$ of Table I. We show in Fig. 5 only those curves that cross half-filling closest to their respective experimental data points. We also show doping curves (small open symbols) for larger values of $U$ and correspondingly larger values of $\epsilon_p$ for LCO and NCCO, thus keeping the nominal gap $\Delta_n = U - \epsilon_p$ constant.

Fig. 6 shows the computed density of states (DoS) at half-filling for each of the three materials studied, with the values of $U$ corresponding to the full circles in Fig. 5. We see that LCO has a large CTG of approximately $3.5 t_{pp}$ (panel (a)), whereas in YBCO this gap is only about $1.0 t_{pp}$ (panel (b)) and in NCCO the CTG has disappeared (panel (c)). The absence of gap at the Fermi level for NCCO was also predicted by a hybrid DFT-DMFT study [16] in the case where long range magnetic order is not allowed, like here.

We do not observe a superconducting dome on the electron doped side in NCCO, rather like the $U = 8$ curve for BSCCO in Fig. 3a, even though there should be one as the material is superconducting. As we mentioned before, DMFT+LDA computations on a similar model showed that by allowing long range magnetic order a gap should open up around the Fermi level at half-filling [16]. We modified the impurity model (A2) in order to allow for antiferromagnetic order, to see wether a CTG would appear for NCCO at $U = 8.0$. To measure the strength of the antiferromagnetic order we defined the order parameter

$$M = \sum_{\mathbf{r}} e^{\mathbf{Q} \cdot \mathbf{r}}(n_{\mathbf{r}\uparrow} - n_{\mathbf{r}\downarrow}) \tag{6}$$

with the antiferromagnetic wave-vector $\mathbf{Q} = (\pi, \pi)$. At

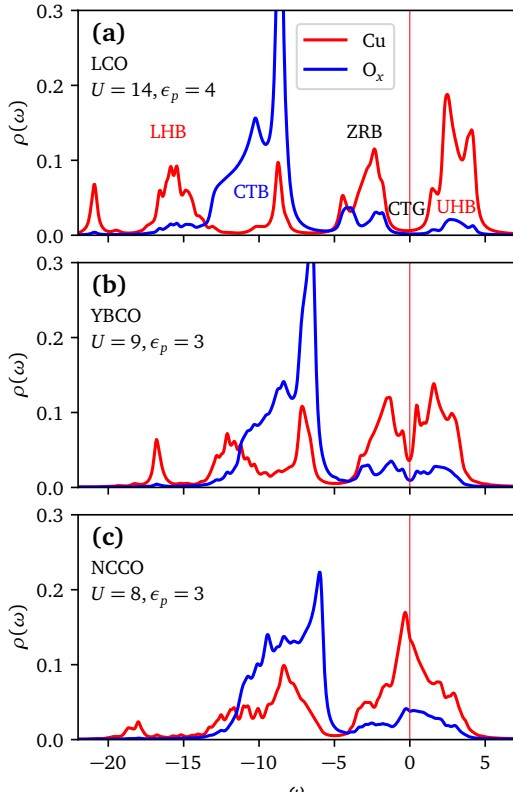

Figure 6. Density of states for the solutions crossing the half-filling axis of Fig. 5 for each of the three cuprate compounds. The Fermi level is the vertical red line. The Cu contribution is shown in red and the contribution of one of the oxygen orbitals in blue (the two oxygens contribute equal amounts).

$(U, \epsilon_p) = (8.0, 3)$ only a weak antiferromagnetic order was observed, which was not strong enough to open a CTG. We then studied the relationship between the value of interaction $U$ and the antiferromagnetic order parameter, as shown in Fig. 7. We notice a discontinuity in the value of $\langle M \rangle$ in between $U \approx 9.06$ and $U \approx 8.90$. This discontinuity is characterized by the opening of an antiferromagnetic order-induced CTG at the Fermi level, as can be seen from Fig. 8. The nature of antiferromagnetism goes from local to itinerant as $U$ is decreased across this discontinuity.

## IV. DISCUSSION AND CONCLUSION

Comparing the hole distribution data from NMR and CDMFT should in principle allow us to set the interaction strength $U$ and the relative band energy $\epsilon_p$ in the Emery model. Knowing band parameters from first-principles calculations, one can produce curves similar to Fig. 2 and approximately identify the right $U - \epsilon_p$ by looking at the intersection of their curves with the half-filling axis and compare it with a NMR measurement on the undoped compound, like in Fig. 5. Ideally, $(n_d, n_p)$ NMR data and the CDMFT results would have the same slope away from half-filling.

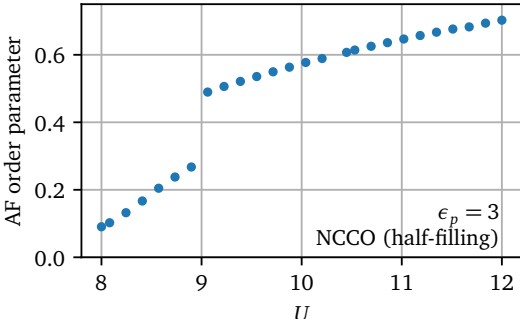

Figure 7. Amplitude of the antiferromagnetic order parameter of NCCO at half-filling in relation to the interaction strength $U$.

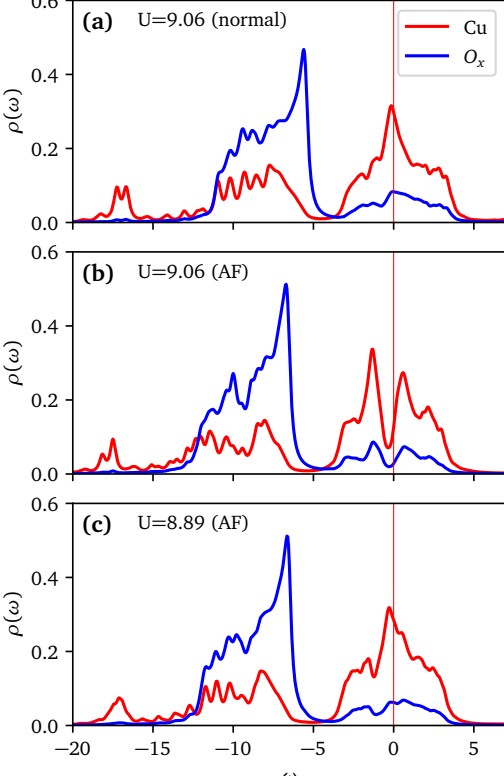

Figure 8. DoS of NCCO in the normal phase in which we allowed for long range antiferromagnetic order. The two values of interaction $U$ shown are the values on the border of the discontinuity in Fig. 7.

From Fig. 5 one sees that this is not exactly the case, although the agreement for YBCO is satisfactory given the dispersion of NMR data.

In LCO, the slope from the 4 NMR data points is smaller than predicted by CDMFT for $\epsilon_p = 4$. This behavior indicates that the O and Cu orbitals are actually less hybridized than the DFT-derived parameters of the Emery model lead us to believe. There could be many causes. Maybe the value of $\epsilon_p$ is actually larger than predicted by DFT. Indeed, our results are closer to the mark if we adopt the more extreme

values $U = 20$ and $\epsilon_p = 10$ (open blue circles in Fig. 5): A larger value of $\epsilon_p$ lowers the hybridization of the Cu and O bands and produces a flatter curve. Maybe also our neglect of the local Coulomb interactions on the oxygens ($U_p$) is to blame, as its importance will grow with hole doping. Finally, this may just be an inherent insufficiency of the Emery model to describe the actual material, which only a treatment of all orbitals could resolve, in the line of Ref. [27].

More serious is the disagreement between the data for NCCO (open red squares on Fig. 5) and the model's predictions on the electron-doped side. The NMR data consists of two points only, but these two points hint at a rather large value of $dn_d/dn_p$, as if the upper Hubbard band contained little oxygen, even though the values of $U$ and $\epsilon_p$ are such that the CTG vanishes.

The value $U - \epsilon_p = 5$ for NCCO, as determined by the position of $(n_p, n_d)$ at half-filling, was not large enough to open a CTG even when allowing long range antiferromagnetic order. Figs 7 and 8 show that a CTG opens up at the Fermi level between $U = 8.90$ and $U = 9.06$ for $\epsilon_p = 3$, leading to a jump in the AF order parameter $\langle M \rangle$. Knowing that NCCO is a superconductor, we should observe a non-negligible value of the SC order parameter $\Psi$ at $U - \epsilon_p = 5$ if it is indeed the right value to describe NCCO in the Emery model, but we don't (not shown). We are clearly underestimating the value of $U - \epsilon_p$ in that case. For hole-doped compounds, which are commonly believed to possess greater $U$ values than their electron-doped counterparts [16, 28], this underestimation is less of a problem since they stand further from the transition between a CTI and a metal. As for the electron-doped cuprates, some argue that they are simply not in the CTI regime [16]. This also points towards a possible (small) disconnect between the computation of densities and that of the SC order parameter, the latter being more affected by the finite size of the cluster since it is less of a local quantity: neglecting inter-cluster components of the anomalous self-energy has more consequences when computing $\Psi$.

Let us point out some deficiencies of the Emery model or of our treatment thereof:

1. In principle one could construct a tight-binding model that fits the DFT band structure perfectly, with a large-enough number of orbitals. The Emery model keeps only three of them, hence some hydbridization with other bands is lost and this will affect the occupation numbers $n_d$ and $n_p$, especially if we venture too far from half-filling, the usual operating point at which the model parameters are derived.

2. In the model itself, we have neglected all interactions but the on-site interaction $U$ on copper orbitals, effectively treating other orbitals in the mean-field approximation. Since the occupation $n_p$ of oxygens is close to 2, this is not such a bad approximation (fluctuations of these interactions are small) but it is less and less valid as hole-doping increases.

3. The value of $\epsilon_p$, as insisted upon above, is strongly renormalized by the interaction, so much so that it is

better to treat it as an adjustable parameter. However, that renormalization is doping dependent, and this was ignored here.

Despite these shortcomings, the Emery model achieves a lot [20]: it reproduces the correlation between superconductivity and superexchange[29], charge transfer gap [30, 31] and oxygen hole doping [22].

To conclude, studying the relation between oxygen occupation $n_p$ and copper occupation $n_d$ can help determine the degree of correlation $U - \epsilon_p$ in the model and set this parameter combination from observations. This works reasonably well for the hole-doped cuprates LSCO ($U - \epsilon_p = 10$) and YBCO ($U - \epsilon_p = 6$), but less so for the electron-doped cuprate NCCO. In the latter case, $U - \epsilon_p$ may be strongly underestimated, or the weaker correlations make the CDMFT approach used here less accurate.

## ACKNOWLEDGMENTS

Fruitful conversations with A.-M. S. Tremblay are gratefully acknowledged. This work has been financially supported by the Natural Sciences and Engineering Research Council of Canada (Grants No. RGPIN-2020-05060) and by the Canada First Research Excellence Fund. Computational resources were provided by the Digital Research Alliance of Canada and Calcul Québec.

## Appendix A: Cluster dynamical mean-field theory

In the appendix we review some technical details of cluster dynamical mean field theory (CDMFT) [32, 33] as applied to our problem. General reviews of CDMFT may be found in Refs [19, 34]. We define an impurity problem consisting of 4 unit cells of the Emery model, which we further decompose into a 4-site, correlated cluster of Cu $d_{x^2-y^2}$ orbitals, plus an 8-site, uncorrelated cluster of O $p$ orbitals. The correlated Cu cluster is augmented by an 8-orbital bath that represents that cluster's environment, and defines an "impurity problem" whose Hamiltonian is

$$H_{\text{imp}} = H_{\text{clus}} + H_{\text{bath}} + H_{\text{hyb}}, \tag{A1}$$

with the following partial Hamiltonians:

$$H_{\text{clus}} = U \sum_{\mathbf{r}} n_{\mathbf{r}\uparrow} n_{\mathbf{r}\downarrow} - \sum_{\mathbf{r},\mathbf{r}',\sigma} t_{\mathbf{r}\mathbf{r}'} c_{\mathbf{r}\sigma}^\dagger c_{\mathbf{r}\sigma} \tag{A2}$$

$$H_{\text{bath}} = \sum_{\alpha,\sigma} \varepsilon_\alpha b_{\alpha\sigma}^\dagger b_{\alpha\sigma} \tag{A3}$$

$$H_{\text{hyb}} = \sum_{\mathbf{r},\alpha,\sigma} \theta_{\mathbf{r}\alpha} (c_{\mathbf{r}\sigma}^\dagger b_{\alpha\sigma} + \text{H.c}) \tag{A4}$$

The index $\mathbf{r}$ stands for site within the cluster and $\alpha$ is the bath orbital index. The $t_{\mathbf{r}\mathbf{r}'}$ are elements of the cluster hopping matrix $\mathbf{t}$, $c_{\mathbf{r}\sigma}$ and $b_{\alpha\sigma}$ are destruction operators on cluster and bath orbitals, respectively, for spin $\sigma = \uparrow, \downarrow$. $\varepsilon_\alpha$ is

the energy of a bath orbital and the $\theta_{\mathbf{r}\alpha}$ are elements of the hybridization matrix that allows electrons to hop between cluster and bath orbitals. The effect of the bath environment on the electron Green function is contained in the hybridization function:

$$\Gamma_{\mathbf{r}\mathbf{r}'}(\omega) = \sum_\alpha \frac{\theta_{\mathbf{r}\alpha} \theta_{\mathbf{r}'\alpha}^*}{\omega - \varepsilon_\alpha} \tag{A5}$$

We can extract the correlated cluster's self-energy $\boldsymbol{\Sigma}_c$ from the Cu orbital Green function obtained via ED through Dyson's equation:

$$\mathbf{G}_c^{-1}(\omega) = \omega - \mathbf{t} - \boldsymbol{\Gamma}(\omega) - \boldsymbol{\Sigma}_c(\omega). \tag{A6}$$

Here $\mathbf{G}_c$ is the exact $8 \times 8$ Green function of the Cu orbitals on the correlated cluster (8 because of spin). The self-energy is solely associated to the Cu cluster since we neglect correlation on the O orbitals. From this self-energy, we construct an approximation to the Green function of the infinite system

$$\mathbf{G}^{-1}(\tilde{\mathbf{k}}, \omega) = \mathbf{G}_0^{-1}(\tilde{\mathbf{k}}, \omega) - \boldsymbol{\Sigma}_c(\omega). \tag{A7}$$

Where $\mathbf{G}_0^{-1}$ is the non-interacting part of the lattice Green function and $\tilde{\mathbf{k}}$ is the reduced wave vector in the Brillouin zone of the super-lattice of repeated clusters. Here $\mathbf{G}_0^{-1}(\tilde{\mathbf{k}})$ is a $24 \times 24$ matrix, since it also contains the contribution of the oxygen orbitals, and $\boldsymbol{\Sigma}_c$ has been padded with zeros from $8 \times 8$ to $24 \times 24$ to include the null contribution of the oxygen orbitals to the self-energy. From this expression, we can in turn calculate the local Green function:

$$\bar{\mathbf{G}}(\omega) = \frac{L}{N} \sum_{\tilde{\mathbf{k}}} \frac{1}{\mathbf{G}_0^{-1}(\tilde{\mathbf{k}}) - \boldsymbol{\Sigma}_c(\omega)}. \tag{A8}$$

The goal of CDMFT is then, through self-consistency, to obtain a local Green function that is equal to the exactly-solved Green function of the cluster: $\mathbf{G}_c(\omega) = \bar{\mathbf{G}}(\omega)$, when restricted to the correlated (Cu) degrees of freedom. With a finite number of bath orbitals, it is impossible for those two functions to coincide perfectly at all frequencies. A distance function is then defined to minimize the difference between the two solutions:

$$d = \sum_{i\omega_n, \nu, \nu'} W_n \left| (\mathbf{G}_c^{-1}(i\omega_n) - \bar{\mathbf{G}}^{-1}(i\omega_n))_{\nu\nu'} \right|^2 \tag{A9}$$

(again, in the above equation, $\bar{\mathbf{G}}^{-1}$ is restricted to the Cu orbitals only). The sum is taken over a suitable set of Matsubara frequencies at some fictitious temperature $1/\beta$, with weights $W_n$ that are usually taken to be constant up to some cutoff frequency $\omega_c$. The distance function is minimized with the help of some minimization method (here we use the Nelder-Mead algorithm [35]), varying the bath parameters, $\varepsilon_\alpha$ and $\theta_{\mu\alpha}$. The newly obtained bath parameters are then used to update the impurity model, and the procedure is iterated until convergence.

Once a self-consistent local Green function has been found, different observables of the system can be computed. Consider for instance a general one-body operator $\hat{S}$:

$$\hat{S} = \sum_{\alpha\beta} s_{\alpha\beta} c_{\alpha}^{\dagger} c_{\beta} \quad . \tag{A10}$$

The ground state expectation value of such an operator is [19]

$$\bar{S} = \int_{-\infty}^{\infty} \frac{d\omega}{2\pi} \frac{d^2\tilde{\mathbf{k}}}{(2\pi)^2} \left\{ \operatorname{tr} \mathbf{s}(\tilde{\mathbf{k}}) \mathbf{G}(\tilde{\mathbf{k}}, \omega) - \frac{\operatorname{tr} \mathbf{s}(\tilde{\mathbf{k}})}{i\omega - p} \right\} \quad , \tag{A11}$$

where convergence is ensured by the subtraction of a pole $p$ located on the positive real axis, if $\operatorname{tr} \mathbf{s} \neq 0$. The expression (A11) is mainly used in this work to compute the electronic (hole) density on Cu and O sites in the unit cell of the model illustrated in Fig. 1.

In this work, we also probe the superconducting state. In that case the impurity model (A2) needs to be upgraded by adding pairing between cluster and bath:

$$H_{\text{anom}} = \sum_{\mathbf{r},\alpha} s_{\mathbf{r}\alpha}(c_{\mathbf{r}\uparrow} b_{\alpha\downarrow} - c_{\mathbf{r}\downarrow} b_{\alpha\uparrow} + \text{H.c}) \quad . \tag{A12}$$

We then use the Nambu formalism, which here amounts to a particle-hole transformation on the spin-down part of the problem and allows to formulate the problem in the same Green function language as in the normal solution. The Nambu off-diagonal blocks of the Green function and self-energy then contain the anomalous parts of these quantities. A superconducting order parameter with $d_{x^2-y^2}$ symmetry is then defined as $\Psi = \langle \hat{\Delta} \rangle / N$ with $N$ the number of unit cells in the lattice and the singlet pairing operator $\hat{\Delta}$ defined as:

$$\hat{\Delta} = \sum_{\langle \mathbf{rr}' \rangle_x} (d_{\mathbf{r}\uparrow} d_{\mathbf{r}'\downarrow} - d_{\mathbf{r}\downarrow} d_{\mathbf{r}'\uparrow}) - \sum_{\langle \mathbf{rr}' \rangle_y} (d_{\mathbf{r}\uparrow} d_{\mathbf{r}'\downarrow} - d_{\mathbf{r}\downarrow} d_{\mathbf{r}'\uparrow}) + \text{H.c} , \tag{A13}$$

with $\langle \mathbf{rr}' \rangle_x$ and $\langle \mathbf{rr}' \rangle_y$ nearest neighbor copper orbitals in the $x$ and $y$ directions, respectively.

Finally, let us point out that the use of an impurity solver based on exact diagonalization (ED), even though limited to small systems sizes, and in particular a discrete bath, has the advantage over quantum Monte Carlo methods that it is not plagued by the fermion sign problem. The latter only gets worse while studying covalent (low values of $\epsilon_p$) compounds compared to ionic compounds. Since cuprates are definitey covalent [15], the use of an ED solver is of great convenience.

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
