# Peer review of "Effect of the Coulomb repulsion and oxygen level on charge distribution and superconductivity in the Emery model for cuprates superconductors"

_SciPost Physics Core_

## Round 1 · Referee Report · Anonymous (Referee 1) · 2025-4-14

Report

The paper analyzed various properties of the Emery model or a three-band model for the cuprates. Based on a comparison of CDMFT calculations of the Cu and O occupations, and NMR experiments, the authors obtain the parameters of the model. They analyze the dependence of the superconducting amplitude on such parameters showing the correlation between superconductivity and superexchange encoded in the Emery model. On the other hand they are unable to reproduce the observed superconductivity in NCCO from the parameters they have extracted from the NMR experiments. This suggests that either CDMFT is unable to capture longer range spatial correlations or that the Emery model neglects important ingredients relevant to NCCO materials or both.

In general I believe that the paper is interesting providing support on the Emery model as a plausible model for the cuprates. It can serve as a platform to improve methods and/or add more terms to the three-band model considered. Hence, I believe the paper deserves publication in Scipost. I enclose some questions/comments that may help to sharpen the message for their consideration.

Comments: - — — - —-

-The results of Fig. 2 namely, that doped holes mostly go into the O orbitals is somehow expected from the relative position of the O and Cu energy levels since e_p>ed? Isn’t this also the idea behind the formation of the Zhang-Rice singlet?

-Wouldn’t the parameter regime U=8-9 shown in Fig. 3 (top) be more consistent with the phase diagram of the cuprates since the electron doped system will show a lower Tc than the hole doped side?

-The CDMFT calculation consists on embedding 4 Cu orbitals + 8 O orbitals in a bath. The bath consists on 8 orbitals connected to the Cu d orbitals. Do the results depend on the size of the bath? For instance, by enlarging (if possible) the bath to three bath orbitals per Cu orbital?

-Interestingly the authors point out: “Note that the data shown in red, with constant nominal gap, corresponds to a single value of J, but a sizeable variation of Ψmax, demonstrating that J is not the only factor at play.” Do the authors have any idea of which other factors may be playing a role in the superconductivity found within their present model?

-The optimal set of parameters for superconductivity seems to be e_p=2, U=8 and x=-0.2 within the present study. Is the general conclusion that for this hole doping range one needs to maximize the superexchange J in order to achieve the strongest superconducting pairing amplitude?

Recommendation

Publish (easily meets expectations and criteria for this Journal; among top 50%)

  • validity: -
  • significance: -
  • originality: -
  • clarity: -
  • formatting: -
  • grammar: -

Author:  David Sénéchal  on 2025-04-23  [id 5409]

(in reply to Report 1 on 2025-04-14)

We cite the original report and comment after each point.

The results of Fig. 2, namely, that doped holes mostly go into the O orbitals is somehow expected from the relative position of the O and Cu energy levels since ep>ed? Isn’t this also the idea behind the formation of the Zhang-Rice singlet?

It is indeed true that the difference in energy between the two orbitals, i.e., ep - ed, leads to the doped holes going to the O orbitals in this parameter regime. But it is also necessary to keep in mind the important relationship between ep and the value of interaction U. As we explained in the paper, increasing the value of U and keeping the nominal gap \Delta_n=U-ep constant will drain the Zhang-Rice band (ZRB) of the Cu states belonging to the lower Hubbard band (LHB), highlighting the fact that the ZRB contains Cu content from both Hubbard bands. But if we went the other way around, by keeping U constant and increasing the value of ep, it would decrease the width of the charge transfer gap (CTG), thus pushing our solution towards the metallic limit instead of the atomic limit. In this case, even though the difference between ep and ed increases, the doped holes would spread more evenly on the Cu and O orbitals, similar to the U=8 and U=8.5 curves of Fig. 2(left), than for lower values of ep.

Wouldn’t the parameter regime U=8-9 shown in Fig. 3 (top) be more consistent with the phase diagram of the cuprates since the electron doped system will show a lower Tc than the hole doped side?

A quick note here: we made a mistake in Fig. 3, the curve labeled U=9 should be labeled U=8.5 instead. Although it is true that Tc is lower for electron-doped cuprates than for their hole-doped counterparts, the parallel between the U=8 and U=8.5 curves of Fig. 3(top) and real materials can't be established. We see that, for these curves, the SC order parameter at half-filling is non-zero, indicating that they do not represent charge-transfer insulators. Thus, the parameter regime U=8-8.5, in this case, is not relevant to actual cuprates.

It is also important to keep in mind that different materials would have different band parameters in the Emery model. In these curves only the value of the interaction U is changed, which does not translate to direct materials comparisons. We also did not allow for long range antiferromagnetic (AF) order in these computations. Since there is a competition between the AF and SC orders, allowing for long range AF would push the superconducting domes away from half-filling, towards higher values of doping.

Finally, we also need to mention that the correspondance between Tc in real materials and the SC order parameter at zero temperature is not a direct one. The SC order parameter is only a proxy for Tc.

The CDMFT calculation consists on embedding 4 Cu orbitals + 8 O orbitals in a bath. The bath consists on 8 orbitals connected to the Cu d orbitals. Do the results depend on the size of the bath? For instance, by enlarging (if possible) the bath to three bath orbitals per Cu orbital?

The 8 bath orbitals are each connected to the 4 Cu orbitals of the lattice. But yes, ideally we would have an infinite number of bath orbitals to have a "continuous" bath environment. Increasing the number of bath orbitals would yield more accurate results but also exponentially increase the computation time and memory (or simply be impossible), as we are using an ED impurity solver. But even with only 8 bath orbitals, it has been verified in other contexts (e.g. Fig. 23 of arXiv :2410.10019, Bacq-Labreuil et al, to appear in PRX) that the hybridization is very well approximated by a discrete bath of 8 orbitals.

Interestingly the authors point out: "Note that the data shown in red, with constant nominal gap, corresponds to a single value of J, but a sizeable variation of \Psi_{max}, demonstrating that J is not the only factor at play". Do the authors have any idea of which other factors may be playing a role in the superconductivity found within their present model?

It seems that when the spectral gap is very small, the relationship between J and the SC order parameter changes drastically (they are no longer proportional to each other). We observe that, in that situation, it is the size of the spectral gap that impacts the order parameter. The smaller the gap, the bigger the order parameter.

The optimal set of parameters for superconductivity seems to be ep=2, U=8 and x=-0.2 within the present study. Is the general conclusion that for this hole doping range one needs to maximize the superexchange J in order to achieve the strongest superconducting pairing amplitude?

Maximizing the superexchange J gives the strongest pairing amplitude in the parameter regime where we see a CTG. When the CTG is closed the relationship between J and the SC order parameter is lost. For a closed gap, let's say at U=8 for BSCCO, J is bigger than for U=8.5 but the maximum SC order parameter is smaller.

---

## Editorial Decision

resubmitted